# Nutrient Availability and Biofilm Polysaccharide Shape the Bacillaene-Dependent Antagonism of *Bacillus subtilis* against *Salmonella* Typhimurium

Eli Podnar,[a] Andi Erega,[a] Tjaša Danevčič,[a] Eva Kovačec,[a] Bram Lories,[b] Hans Steenackers,[b] Ines Mandic-Mulec[a,c]

[a]Department of Microbiology, Biotechnical Faculty, University of Ljubljana, Ljubljana, Slovenia
[b]Department of Microbial and Molecular Systems, Centre of Microbial and Plant Genetics, KU Leuven, Leuven, Belgium
[c]Chair of Microprocess Engineering and Technology (COMPETE), University of Ljubljana, Ljubljana, Slovenia

**ABSTRACT** *Salmonella enterica* is one of the most common foodborne pathogens and, due to the spread of antibiotic resistance, new antimicrobial strategies are urgently needed to control it. In this study, we explored the probiotic potential of *Bacillus subtilis* PS-216 and elucidated the mechanisms that underlie the interactions between this soil isolate and the model pathogenic strain *S.* Typhimurium SL1344. The results reveal that *B. subtilis* PS-216 inhibits the growth and biofilm formation of *S.* Typhimurium through the production of the *pks* cluster-dependent polyketide bacillaene. The presence of *S.* Typhimurium enhanced the activity of the $P_{pksC}$ promoter that controls bacillaene production, suggesting that *B. subtilis* senses and responds to *Salmonella*. The level of *Salmonella* inhibition, overall $P_{pksC}$ activity, and $P_{pksC}$ induction by *Salmonella* were all higher in nutrient-rich conditions than in nutrient-depleted conditions. Although eliminating the extracellular polysaccharide production of *B. subtilis* via deletion of the *epsA-O* operon had no significant effect on inhibitory activity against *Salmonella* in nutrient-rich conditions, this deletion mutant showed an enhanced antagonism against *Salmonella* in nutrient-depleted conditions, revealing an intricate relationship between exopolysaccharide production, nutrient availability, and bacillaene synthesis. Overall, this work provides evidence on the regulatory role of nutrient availability, sensing of the competitor, and EpsA-O polysaccharide in the social outcome of bacillaene-dependent competition between *B. subtilis* and *S.* Typhimurium.

**IMPORTANCE** Probiotic bacteria represent an alternative for controlling foodborne disease caused by *Salmonella enterica*, which constitutes a serious concern during food production due to its antibiotic resistance and resilience to environmental stress. *Bacillus subtilis* is gaining popularity as a probiotic, but its behavior in biofilms with pathogens such as *Salmonella* remains to be elucidated. Here, we show that the antagonism of *B. subtilis* is mediated by the polyketide bacillaene and that the production of bacillaene is a highly dynamic trait which depends on environmental factors such as nutrient availability and the presence of competitors. Moreover, the production of extracellular polysaccharides by *B. subtilis* further alters the influence of these factors. Hence, this work highlights the inhibitory effect of *B. subtilis*, which is condition-dependent, and the importance of evaluating probiotic strains under conditions relevant to the intended use.

**KEYWORDS** *Bacillus subtilis*, *Salmonella enterica*, probiotics, enteric pathogen, biofilm, polysaccharides, microbial competition, nutrients

Address correspondence to Ines Mandic-Mulec, ines.mandicmulec@bf.uni-lj.si.

The authors declare no conflict of interest.

*S*almonella enterica is one of the most prevalent foodborne enteropathogens in the world, causing about 1.35 million illnesses per year in the United States alone (https://www.cdc.gov/salmonella/index.html). It can be transmitted by consumption of contaminated food or water, via an infected animal, or human-to-human (1, 2). The formation of biofilms

allows *Salmonella* to persist in harsh environments and provides protection against common antimicrobial compounds (3). In addition, high levels of antibiotic resistance further complicate the eradication of *Salmonella* contaminations and infections (1). Therefore, novel solutions are urgently required and the use of probiotic bacteria to prevent *Salmonella* contaminations and infections is being explored (4–6).

In recent decades, *Bacillus subtilis* has gained popularity as a probiotic, and it is increasingly used in probiotic preparations (7, 8). While *Lactobacilli* and *Bifidobacteria* strains remain the most commonly used probiotics to combat *Salmonella*, *B. subtilis* also has the potential to be an effective probiotic against *Salmonella*, as it was shown to prevent *Salmonella* invasion in intestinal epithelial cells (9, 10) and protect chickens (11) from *Salmonella* infections, thus reducing transmission to humans. However, the underlying mechanisms which modulate the interaction between *B. subtilis* and *Salmonella* remain largely unknown.

*B. subtilis* is known to produce a broad spectrum of chemically diverse secondary metabolites (12, 13). Most of these compounds are antimicrobial peptides with different roles. For example, *B. subtilis* produces the lipopeptide surfactin, which shows antimicrobial activity against a variety of bacteria (14) and inhibits adhesion of *Salmonella* to surfaces (15). In addition, the non-ribosomal peptide/polyketide bacillaene inhibits bacterial protein synthesis (16). This compound is produced by the enzymatic megacomplex non-ribosomal peptide synthetases (NRPS) and polyketide synthases (PKS), which are encoded on a biosynthetic *pks* gene cluster (17). Bacillaene acts as an antagonist in various interspecies interactions (17–20) and is suggested to be an important player in *B. subtilis*-based probiotics (21, 22). It has been shown that bacillaene decreases biofilm formation in Gram-negative foodborne pathogen *Campylobacter jejuni* (21) and inhibits both growth and antibiotic synthesis in different *Streptomyces* strains (17, 19). However, how bacillaene production influences the interaction of *B. subtilis* with *Salmonella* has not yet been explored.

In addition to producing a diverse set of antimicrobials, *B. subtilis* is also a ubiquitous microorganism that inhabits many different ecosystems (23), including the gastrointestinal tracts of humans and animals (24–26). Moreover, *B. subtilis* can form biofilms both as a pellicle on an air-liquid interface (27, 28) and as a submerged biofilm attached to a solid surface (28–30). Therefore, *B. subtilis* has the potential to colonize the same niches as *Salmonella*, further enhancing the effect of its secreted antimicrobial compounds.

This work investigates the mechanisms that shape the interactions between the model biofilm former and potential probiotic strain *B. subtilis* PS-216 (31–33) and the model pathogenic strain *S.* Typhimurium SL1344 (34) in a static biofilm model. We explored this interaction at different nutrient concentrations because it has been shown that nutrient availability influences various factors important for competition (35), including the production of antimicrobial compounds (36, 37) and biofilm formation (38–40). We revealed that *B. subtilis* inhibits growth and adhesion to surfaces in *S.* Typhimurium in a nutrient concentration-dependent manner via the production of bacillaene. Moreover, biofilm formation and the presence of *Salmonella* further alters the production of bacillaene.

## RESULTS

**Competition between *B. subtilis* and *S.* Typhimurium in static culture.** We aimed to test the competition between the pathogen *S.* Typhimurium SL1344 and the potential probiotic strain *B. subtilis* PS-216 in mixed-species biofilms. We tested the effect of nutrient availability on competition in static coculture by utilizing nutrient-rich (tryptic soy broth [TSB]), nutrient-restricted (1/5 TSB), and nutrient-depleted (1/20 TSB) media. Both species were labeled with an antibiotic resistance gene, which allowed their quantification by CFU counting, directly upon mixing at a 1:1 ratio and after 24 h of incubation in TSB medium. The results showed that *B. subtilis* inhibited *S.* Typhimurium growth at 1.3 log in nutrient-rich conditions, while no growth inhibition of *B. subtilis* was observed in coculture (Fig. 1). Nutrient-restricted conditions reduced the growth of both species as well as the antagonistic potential of *B. subtilis*. Under nutrient-

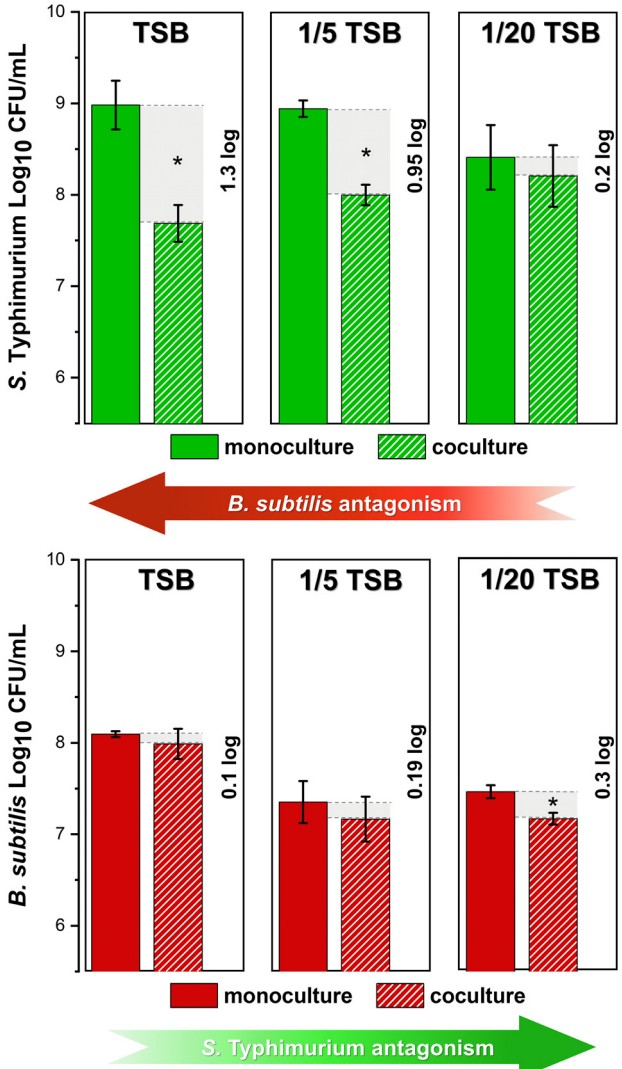

**FIG 1** The influence of nutrients on cell counts of *Bacillus subtilis* BM1097 and *Salmonella* Typhimurium SL1344 GFP in monocultures and cocultures. Bacterial cells were grown in tryptic soy broth (TSB), 1/5 TSB, and 1/20 TSB media and CFU/mL values were determined after 24 h of static incubation at 37°C. Data sets represent means and standard deviation of three biological replicates. Statistically significant results ($P < 0.05$) were determined using Student's *t* test and are indicated with an asterisk (*).

depleted conditions *S.* Typhimurium fully escaped growth inhibition and even significantly inhibited the growth of *B. subtilis* (Fig. 1).

**Spatial distribution of *Salmonella* and *Bacillus*.** Since the spatial organization of a community can strongly influence the level of competition (35), we determined the cell distribution of a fluorescently labeled *B. subtilis* PS-216 WT (mKate2) and an *S.* Typhimurium SL1344 (GFP) strain by confocal laser scanning microscopy (CLSM) using the whole-volume 3D microscopy approach. Moreover, the effect of medium dilution on *B. subtilis* biofilm architecture has not yet been studied. In nutrient-rich monoculture conditions, *S.* Typhimurium formed submerged biofilms, whereas the *B. subtilis* strain formed both a submerged biofilm and a pellicle at the air-liquid interface. The spatial organization did not change in coculture conditions, and the submerged *B. subtilis* biofilm was localized just above the *Salmonella* biofilm (Fig. 2A). The localization of *S.* Typhimurium did not change in media with lower nutrient concentrations; however, *B. subtilis* no longer formed a pellicle and only produced a submerged biofilm, which in coculture was again localized just above *S.* Typhimurium (Fig. 2B, C). Macroscopic biofilm images support these observations (Fig. S1 in the supplemental material).

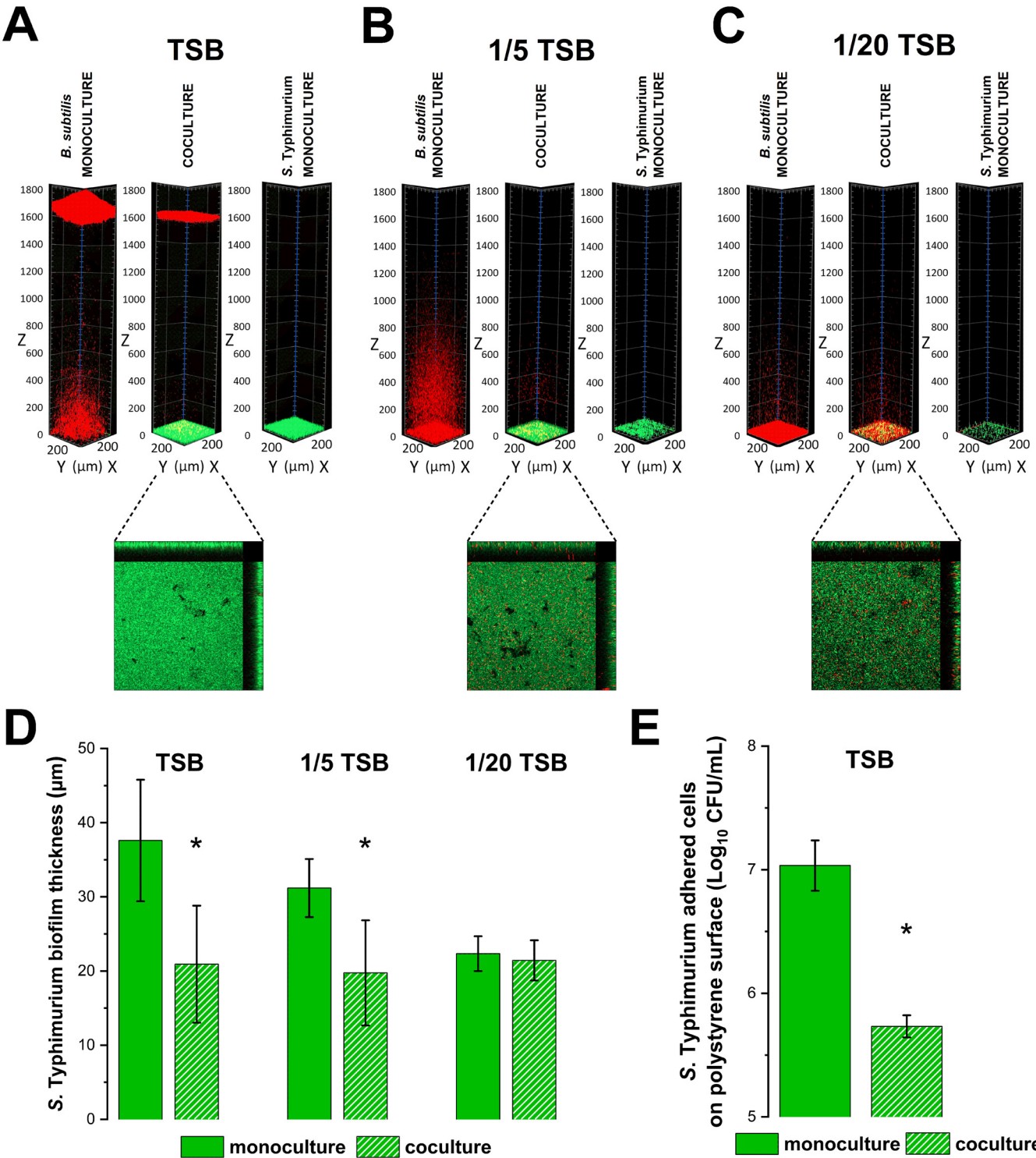

**FIG 2** Spatial distribution of *B. subtilis* BM1097 (red) and *S.* Typhimurium SL1344 GFP (green) and thickness of *S.* Typhimurium SL1344 GFP biofilm in different growth media. (A to C) 3D view of *B. subtilis* wild-type (WT) strain BM1097 and *S.* Typhimurium SL1344 GFP monocultures and cocultures in TSB (A), 1/5 TSB (B), and 1/20 TSB medium (C) after 24 h of static incubation at 37°C. Only orthogonal views of *B. subtilis* and *S.* Typhimurium cocultures in three different media are shown. The middle image in the orthogonal view represents the first slice (319 $\mu$m × 319 $\mu$m), which was measured from the bottom of the well. Side and top images in the orthogonal view represent the y to z and x to z stacks, respectively, with z-stack 50 $\mu$m in size. (D) Thickness of *S.* Typhimurium SL1344 GFP biofilm in monocultures and cocultures in TSB, 1/5 TSB, and 1/20 TSB media after 24 h of static incubation at 37°C. (E) Adhesion of *S.* Typhimurium SL1344 GFP on polystyrene surface after 24 h of static incubation in monoculture and coculture with *B. subtilis* WT strain BM1097 at 37°C in TSB medium. Data sets represent means and standard deviation of three biological replicates. A Student's *t* test was used to determine statistically significant differences ($P < 0.05$) to monoculture, which are marked with an asterisk (*).

Moreover, in nutrient-depleted conditions, the interspecies mixing was more intimate, and a few *B. subtilis* clusters were even interspersed between *S.* Typhimurium cell patches, as seen in the orthogonal view of the biofilm representing the first 50 $\mu$m of the z-stack (Fig. 2C). This increased mixing is associated with a lower degree of inhibition by *B. subtilis* and increased inhibition by *S.* Typhimurium (Fig. 1). Consistently, the *S.* Typhimurium biofilm was most robust in the nutrient-rich environment, with an average thickness of 37 $\mu$m as shown by confocal microscopy (Fig. 2D). *S.* Typhimurium biofilm thickness decreased in coculture. The greatest effect of interspecies interactions was observed in nutrient-rich medium, whereas in nutrient-depleted conditions, the reduction of *S.* Typhimurium biofilm thickness in coculture compared to that in monoculture was not significant (Fig. 2D). These results are in line with the reduced inhibition of *S.* Typhimurium under nutrient-depleted conditions (Fig. 1).

Adhesion to surfaces is the first step in surface colonization and the formation of a mature, resistant biofilm (41). By controlling adhesion, we can control the maturation and dispersal of biofilms. To further test the potential of *B. subtilis* to disrupt formation of *S.* Typhimurium biofilms, we cocultured both strains in nutrient-rich conditions and determined the adhesion of *S.* Typhimurium to the polystyrene surface. This assay confirmed that the *B. subtilis* strain was able to significantly reduce *S.* Typhimurium adhesion, probably due to growth inhibition (Fig. 2E).

**Antagonistic behavior in *B. subtilis* is due to bacillaene production.** To test whether the production of antimicrobial compounds underlies the strong inhibition of *Salmonella* under nutrient-rich conditions, we compared levels of *S.* Typhimurium inhibition in coculture between *B. subtilis* strains PS-216 and 168. *B. subtilis* PS-216 is a natural soil isolate (33), while *B. subtilis* 168 is a laboratory strain defective in the *sfp* gene responsible for synthesis of different antimicrobial compounds (42–44). *B. subtilis* 168 did not inhibit the growth of *S.* Typhimurium (Fig. 3A), indicating that the antagonism of *B. subtilis* PS-216 is indeed due to the production of antimicrobial compounds. We then tested several PS-216 mutants defective in either surfactin ($\Delta srfAA$), plipastatin ($\Delta ppsB$), or bacilysin ($\Delta bacA$). However, all tested mutants showed similar levels of *Salmonella* inhibition as the *B. subtilis* PS-216 WT (wild-type) strain, suggesting that these antimicrobial compounds do not contribute to *S.* Typhimurium growth inhibition (Fig. 3A). In contrast, the *B. subtilis* PS-216 $\Delta pks$ mutant, which does not produce bacillaene, completely lost its inhibitory effect against the pathogen (Fig. 3A) and consequently its ability to interfere with the adhesion of *Salmonella* to the polystyrene surface (Fig. 3B). Moreover, when the $\Delta pksC$ deletion was complemented by the *pksC* gene integrated into the *sacA* locus, the mutant was restored to the wild-type phenotype and could again inhibit *S.* Typhimurium (Fig. S2). This confirms that bacillaene is the major antagonist of *B. subtilis* against *S.* Typhimurium in nutrient-rich conditions.

Next, we explored whether a difference in bacillaene production underlies the reduced inhibition by *B. subtilis* in nutrient-depleted conditions. Here, we measured the expression of the *pks* operon via a P$_{pksC}$-*yfp* reporter fusion in nutrient-rich and nutrient-depleted conditions, since it has been previously established that activity of the P$_{pksC}$ promoter correlates well with bacillaene synthesis (45, 46). In addition, because it has been shown that bacteria can alter the production of antimicrobials in the presence of competitors (45, 47, 48), we compared the expression between monoculture and coculture conditions (Fig. 3C). Overall, the *pks* operon showed a much stronger activity in nutrient-rich than in nutrient-depleted conditions, possibly explaining the lack of *Salmonella* inhibition in nutrient-depleted conditions. In nutrient-rich conditions, the level of P$_{pksC}$-*yfp* activity was initially comparable between monoculture and coculture. However, at the 8- to 10-h time period, a dramatic increase in P$_{pksC}$-*yfp* activity was observed in coculture with *S.* Typhimurium, which was absent in the monoculture. Consistently, we also observed the increase in P$_{pksC}$-*yfp* promoter activity in cocultures at the 16-h time point using flow cytometry; however, P$_{pksC}$-*yfp* activity was no longer increased at 24 h. Possibly, partial lysis of the *Bacillus* population in stationary phase (49, 50) resulted in the release of yellow fluorescent protein (YFP) into the surrounding medium, which could still be detected by the microplate reader method but not by

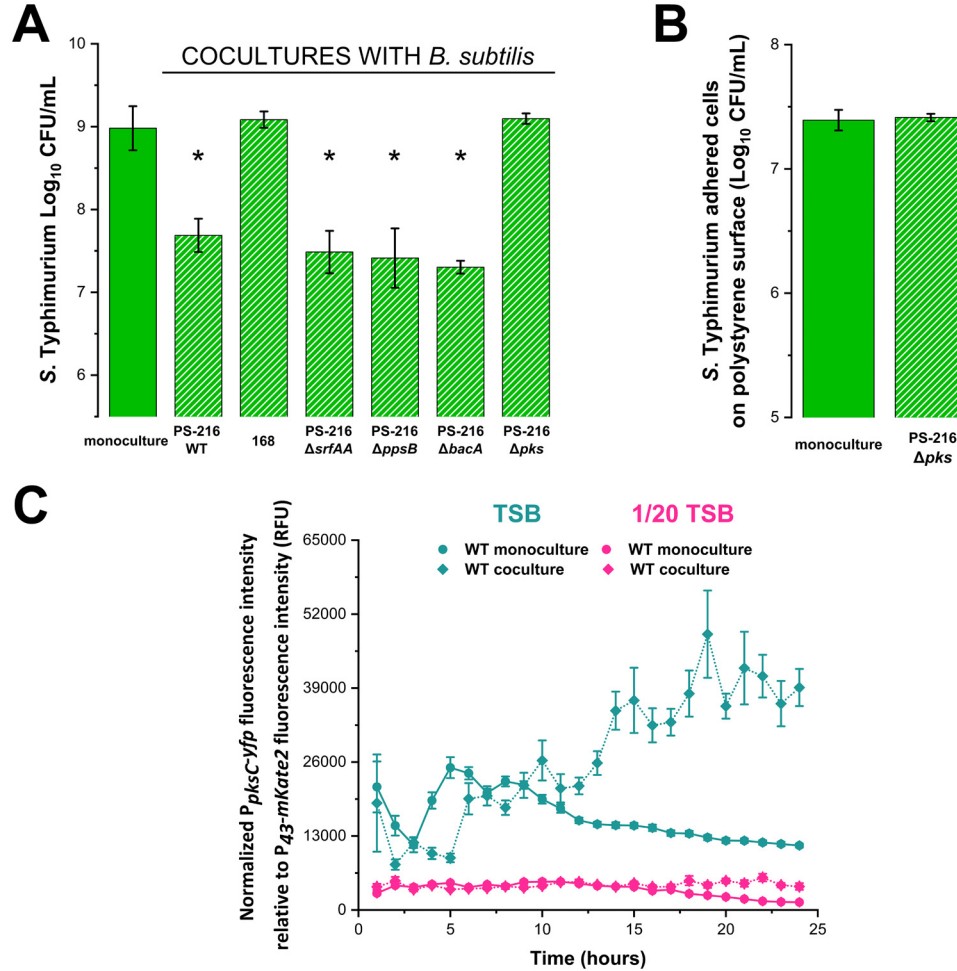

**FIG 3** The role of bacillaene in competition with *Salmonella*. (A) Effect of different *B. subtilis* strains on *S.* Typhimurium SL1344 GFP growth in TSB medium after 24 h of static incubation at 37°C. (B) Adhesion of *S.* Typhimurium SL1344 GFP on polystyrene surface after 24 h of static cocultivation with the *B. subtilis* PS-216 Δ*pks* mutant (BM1957) at 37°C in TSB medium. Data sets represent mean values with the standard deviation of three biological replicates. A Student's *t* test was used to determine statistically significant differences ($P < 0.05$) to monoculture, which are marked with an asterisk (*). (C) Transcriptional activity of P$_{pksC}$-*yfp* promoter of the *B. subtilis* BM1884 strain in monoculture and coculture with *S.* Typhimurium SL1344. Measurements of P$_{pksC}$-*yfp* promoter activities were performed in TSB and 1/20 TSB media every half-hour (for clarity, only data points measured every hour are shown). Results are presented as relative units, as described in the Methods section, using a constitutively expressed P$_{43}$-*mKate2* promoter as a proxy for *B. subtilis* biomass.

flow cytometry (Fig. S3). In nutrient-depleted conditions, P$_{pksC}$-*yfp* promoter activity dropped after 12 h in monoculture conditions, whereas in coculture activity remained constant (Fig. 3C). Similarly, P$_{pksC}$-*yfp* promoter activity remained uninduced when the promoter activity under monoculture and coculture conditions was monitored by flow cytometry (Fig. S3). Interestingly, conditioned medium isolated from *S.* Typhimurium monoculture or coculture grown in nutrient-rich conditions did not induce P$_{pksC}$-*yfp* promoter activity (Fig. S4). These results underscore the importance of interspecies interactions and nutrient availability in regulating bacillaene production.

**B. subtilis mutant lacking polysaccharide matrix antagonizes S. Typhimurium biofilm even under nutrient-depleted conditions.** It was previously reported that biofilm formation could alter the production of bacillaene and mediate the outcome of interspecies interactions (46). The extracellular matrix of the *B. subtilis* biofilm is composed of polysaccharides, proteins, and nucleic acids (27, 40, 51). The polysaccharide component of the extracellular matrix contributes to pellicle formation and biofilm structure and is synthesized by proteins made from the *epsA-O* operon (52). To determine how the production of polysaccharides influences competition with *Salmonella*, we utilized a *B. subtilis* PS-216

Δ*epsA-O* mutant which does not produce extracellular polysaccharides and is defective in pellicle formation (32, 53). This PS-216 Δ*epsA-O* mutant showed deficient pellicle formation in nutrient-rich conditions and only formed a submerged biofilm at the bottom of the well in both monoculture and coculture, resulting in increased mixing with the pathogen (Fig. 4A, Fig. S1). Next, growth inhibition of *S.* Typhimurium by *B. subtilis* Δ*epsA-O* mutant was tested in nutrient-rich and nutrient-depleted media. Although the Δ*epsA-O* mutant had a slightly lower inhibitory effect on *S.* Typhimurium growth than the WT strain in the nutrient-rich medium, it was a significantly better inhibitor of *Salmonella* under nutrient-depleted conditions (Fig. 4B). Inhibition by the PS-216 Δ*epsA-O* mutant was also dependent on bacillaene, as the *B. subtilis* PS-216 Δ*epsA-O*Δ*pks* double mutant completely lost its antagonism in both nutrient-rich and nutrient-depleted conditions (Fig. 4B).

The matrix deficiency of the *B. subtilis* Δ*epsA-O* mutant could potentially confer a metabolic advantage to *B. subtilis* in nutrient-depleted conditions and consequently increase its competitive strength with *Salmonella*. Therefore, we determined the growth rates of the *B. subtilis* Δ*epsA-O* mutant and WT strain. However, there was no difference in growth rates between these two strains in either nutrient-rich or nutrient-depleted conditions (Fig. S5). Thus, increased growth of the *B. subtilis* Δ*epsA-O* mutant could not explain the enhanced inhibition of *Salmonella* in nutrient-depleted conditions.

Subsequently, we compared bacillaene production between the *B. subtilis* WT strain and the Δ*epsA-O* mutant by monitoring the activity of the P$_{pksC}$-*yfp* reporter fusion. Overall, the Δ*epsA-O* mutant showed slightly higher levels of P$_{pksC}$ activity in nutrient-depleted conditions than the WT strain. The Δ*epsA-O* mutant did not significantly induce P$_{pksC}$ activity in the presence of *Salmonella* in either nutrient-rich or nutrient-depleted conditions. After 15 h in nutrient-depleted conditions, the P$_{pksC}$ activity in monoculture conditions strongly declined, whereas it remained stable in coculture (Fig. 4C). However, these observations could not be validated by single-cell measurements (Fig. S6). Nevertheless, the fact that the Δ*epsA-O*Δ*pks* double mutant behaves similarly to the Δ*pks* single mutant in terms of *Salmonella* inhibition still suggests that the effect of eliminating the extracellular polysaccharide is at least partially mediated through bacillaene. In addition, other factors such as delayed sporulation (Fig. S7) in the Δ*epsA-O* mutant can further contribute to the mutant's improved inhibitory efficacy of *Salmonella* under nutrient-depleted conditions.

## DISCUSSION

*Salmonella* is a particularly problematic enteropathogen due to its high levels of antibiotic resistance (54) and its ability to form biofilms, resulting in persistent contaminations in the food industry (55–57). Probiotics such as *B. subtilis* strains are a promising alternative to combat this pathogen and limit the spread of resistant variants (4). Most of the literature dealing with *B. subtilis*-*Salmonella* interactions is conducted in broilers and focuses primarily on the broilers' growth effects and changes in the fecal microbiota (11, 58–62). We have shown that *B. subtilis* PS-216 antagonism is mediated by the polyketide antibiotic bacillaene, as PS-216 which lacks the *pks* operon, responsible for bacillaene synthesis, completely loses its ability to inhibit *S.* Typhimurium growth and adhesion to polystyrene surfaces. This diffusible polyketide antibiotic (12, 17, 63) inhibits the growth of various bacteria when purified or present in spent media (13). However, its influence on competition in a mixed-species biofilm has been less studied (17, 19–21, 46) and has not yet been addressed for *B. subtilis*-*Salmonella* interactions.

In addition, we observed that bacillaene production, and hence *Salmonella* inhibition, strongly depends on nutrient availability. Nutrient-depleted conditions failed to induce the P$_{pksC}$ promoter and decreased the antagonism of *B. subtilis*, despite improved spatial mixing within the submerged biofilm. A decrease in antagonism under low nutrient conditions is unexpected because the synthesis of bacillaene is dependent on the transcription factors Spo0A and CodY, which are both active under nutrient limitation (37, 64). Interestingly, in nutrient-depleted conditions, the Δ*epsA-O* mutant deficient in biofilm formation was a stronger inhibitor of *Salmonella* than the WT strain. These results indicate an intricate interplay between biofilm formation, nutrient availability, and the inhibition of *S.* Typhimurium growth

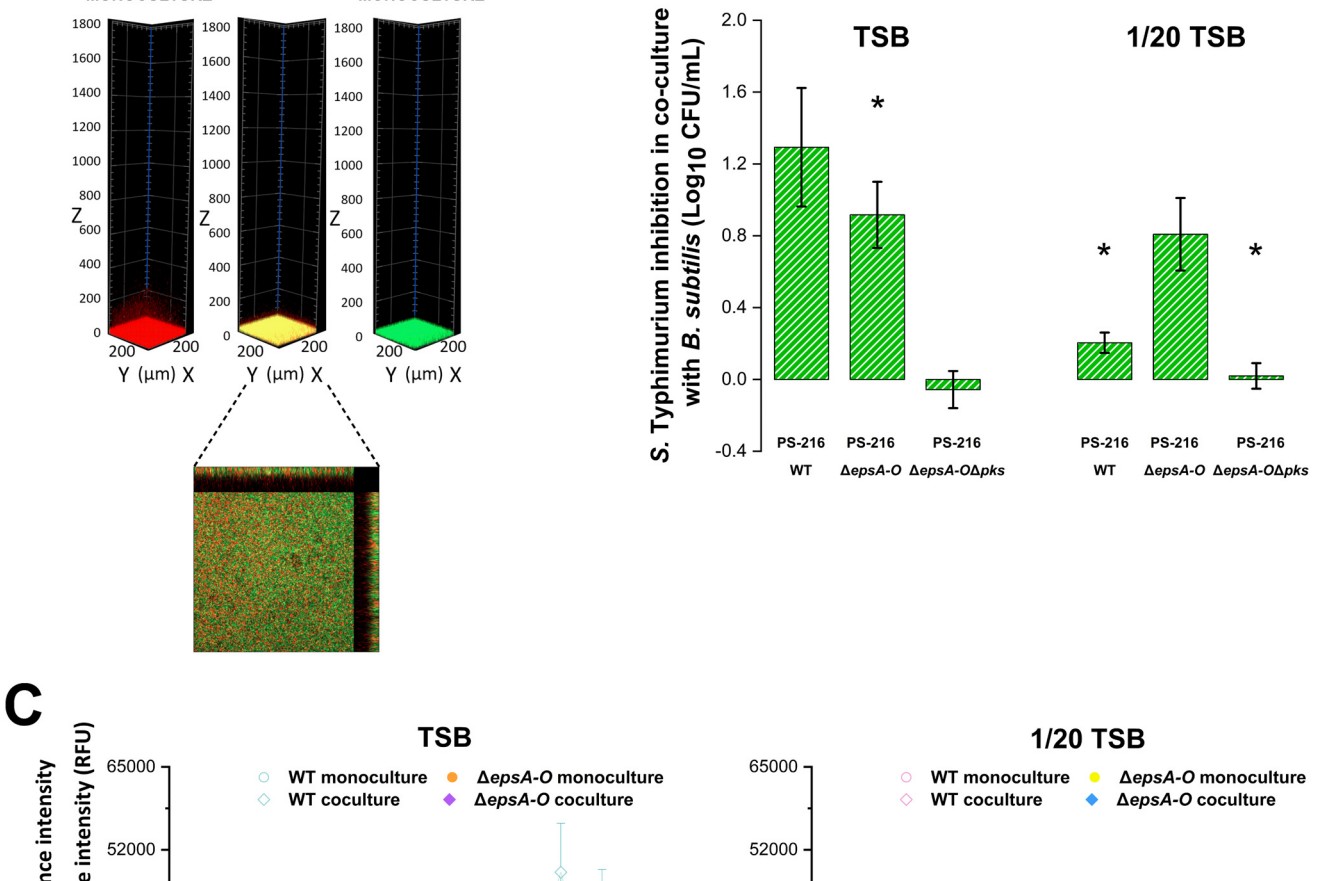

**FIG 4** The influence of biofilm matrix deficiency on bacillaene-mediated antagonism against *S.* Typhimurium. (A) 3D view of monoculture and cocultures of *B. subtilis* PS-216 Δ*epsA-O* mutant (BM1310) (red) and *S.* Typhimurium SL1344 GFP (green) and orthogonal view of coculture in TSB medium after 24 h of static incubation at 37°C. In the orthogonal view, the middle image shows the first slice (319 $\mu$m × 319 $\mu$m) measured from the bottom of the well. In the orthogonal view, side and top images represent the y to z and x to z stacks, respectively, with z-stack 50 $\mu$m in size. A similar spatial distribution was observed in 1/20 TSB medium (data not shown). (B) Growth inhibition of *S.* Typhimurium SL1344 GFP in coculture with the *B. subtilis* PS-216 WT strain (BM1097), *B. subtilis* PS-216 Δ*epsA-O* mutant (BM1310), and *B. subtilis* PS-216 Δ*epsA-O* Δ*pks* double mutant (BM1906). Measurements of CFU/mL were performed after 24 h of coincubation in static conditions in TSB and 1/20 TSB media at 37°C. Data sets represent mean values with the standard deviation of three biological replicates. A Student's *t* test was used to determine statistically significant differences (*, $P < 0.05$) to the *B. subtilis* PS-216 Δ*epsA-O* mutant (BM1310). (C) Transcriptional activity of the P*pksC*–*yfp* promoter of the *B. subtilis* BM1884 and BM1901 strains. Measurements of the P*pksC*–*yfp* promoter activity were performed in TSB and 1/20 TSB media every half-hour (for clarity, only data points measured every hour are shown). Results are presented as relative units, as described in Methods, and using the constitutively expressed P*43*-*mKate2* promoter as a proxy for *B. subtilis* biomass.

by *B. subtilis*. As *B. subtilis* matrix mutants show a delay in sporulation (65, 66), it is possible that the Δ*epsA-O* mutant preserves the ability to fight the competitor instead of entering the dormant state, enhancing inhibition of *Salmonella*. However, further experiments are required to verify this hypothesis.

Moreover, we showed that *B. subtilis* upregulates the production of bacillaene in coculture with *Salmonella*. Several mechanisms for detecting competitors and inducing a competitive response may play a role. First, the competition-sensing hypothesis states that bacteria can use their stress response systems to detect damage caused by direct competition via antibiotics and by indirect competition for nutrients (36). However, we observed a higher level of bacillaene induction in nutrient-rich conditions, an environment associated with reduced competition for nutrients, and the lowest level of *B. subtilis* inhibition by *Salmonella* under low-nutrient conditions. There is thus no indication that competition sensing induces bacillaene production in this interaction. Alternatively, bacteria can use the detection of specific molecules as cues for the presence of competitors. It has recently been shown that *B. subtilis* can sense the presence of another *Bacillus* species via peptidoglycan fragments (45). Possibly, fragments released from killed *Salmonella* cells could also induce bacillaene production. This could explain the difference in induction in coculture between nutrient-rich and nutrient-depleted conditions. The higher initial level of bacillaene in nutrient-rich conditions could be sufficient to kill a part of the *Salmonella* population, resulting in the further induction of bacillaene production and the establishment of a positive feedback loop. Finally, bacteria can also eavesdrop on the quorum-sensing signal molecules produced by other bacteria. *Salmonella* Typhimurium is known to secrete the signaling molecule autoinducer-2 (67). Various other species can produce and detect this signaling molecule, including *B. subtilis*. However, no link between autoinducer-2 and bacillaene production has been described so far.

To conclude, although the exact mechanisms by which *B. subtilis* detects the presence of *Salmonella* and upregulates the production of bacillaene remains unknown, we showed here that *B. subtilis* can inhibit growth and biofilm formation in *Salmonella* via the polyketide bacillaene. We further showed that bacillaene synthesis depends on (i) nutrient availability, (ii) polysaccharide matrix production, and (iii) the presence of competitors, highlighting the importance of evaluating the inhibitory effect of probiotic strains on pathogens under conditions that are relevant for the envisioned application.

## MATERIALS AND METHODS

**Bacterial strains, strain construction, and growth conditions.** In this study, *B. subtilis* PS-216- and *B. subtilis* 168-derived strains were used, labeled with a red fluorescent protein, mKate2, the gene for which is linked to a constitutive promoter $P_{hyspank}$ or $P_{43}$ integrated in two different loci, namely, *amyE::* $P_{hyspank}$-*mKate2* and *sacA::*$P_{43}$-*mKate2* (Table 1). The *B. subtilis* mutant strains were obtained by a standard transformation protocol in which the *B. subtilis* PS-216 or 168 recipient strains were transformed by added DNA (68). Briefly, the recipient strains were grown in modified competence (MC) medium at 37°C shaken at 200 rpm for 5 h, and donor DNA (plasmid or PCR product) was added to the cultures. After 1.5 h of culture incubation under the same incubation conditions, transformants were plated on LB agar plates with the appropriate antibiotics: 10 $\mu$g/mL chloramphenicol (Cm), 50 $\mu$g/mL kanamycin (Kn), 10 $\mu$g/mL tetracycline (Tc), 100 $\mu$g/mL or spectinomycin (Sp). The *B. subtilis* PS-216 Δ*pks::spec* mutant was constructed by amplifying the spectinomycin-inactivated *pks* gene cluster from the genomic DNA of *B. subtilis* PSK0178 using the pksX1 and pksX4 primer pair (Table S1) (17). The PCR product was then transformed in different recipient strains such as *B. subtilis* PS-216 WT, *B. subtilis* PS-216 *amyE::*$P_{hyspank}$-*mKate2* (BM1097), and *B. subtilis* PS-216 *epsA-O::tet amyE::*$P_{hyspank}$-*mKate2* (BM1310), producing the *B. subtilis* mutant strains BM1875, BM1876, and BM1906, respectively. The *B. subtilis* PS-216 Δ*pksC* mutant strain was generated by amplifying the erythromycin-inactivated *pksC* gene from the genomic DNA of *B. subtilis* BKE17100 using the 5pL-pksC and 3pR-pksC primers (69) (Table S1). The PCR product of the erythromycin-inactivated *pksC* gene was then transformed into the recipient strain of *B. subtilis* PS-216 WT, resulting in *B. subtilis* BM1957. The *B. subtilis pksC* complementation mutant was generated by amplifying and the *pksC* gene from the genomic DNA of *B. subtilis* PS-216 WT using the primer pair pksC compl-F (HindIII) and pksC compl-R (BamHI) (Table S1). The PCR product was digested with HindIII and BamHI restriction enzymes and ligated into the previously digested plasmid pSac-Cm (70), which allows integration of a gene into the *sacA* locus. The integration was then verified by sequencing. The obtained plasmid pEM1112 (Table S2) was then transformed into BM1957 to create the *B. subtilis* BM1959 strain.

To assay bacillaene biosynthetic gene expression, we generated the plasmid pEM1108 (Table S2) carrying a $P_{pksC}$-*yfp* reporter fusion. Briefly, the $P_{pksC}$ promoter region was PCR-amplified from *B. subtilis* PS-216 WT using the pC-F(EcoRI) and pC-R(HindIII) primer pair (Table S1). The PCR product was then digested with EcoRI and HindIII restriction enzymes and ligated into the previously digested plasmid pKM3 (71) to obtain the plasmid pEM1108 (Table S2). The plasmid pKM3 was digested with EcoRI and HindIII restriction enzymes to remove the $P_{spoIIQ}$ promoter region from the original vector. Plasmid pEM1108 was further transformed into *B. subtilis* PS-216 *sacA::*$P_{43}$-*mKate2* (BM1629) and PS-216 *epsA-O:: tet* (BM1070) strains, producing the strains *B. subtilis* BM1884 and BM1899, respectively. The latter strain was then transformed with plasmid pMS17 (32) with a kanamycin resistance cassette to generate the *B.*

**TABLE 1** Strains used in this study[a]

| Strain name | Background | Genotype | Reference |
|---|---|---|---|
| *Bacillus subtilis* | | | |
| PS-216 | NA | WT | 33 |
| BM1097 | PS-216 | *amyE*::P$_{hyspank}$-*mKate2* (Cm) | 74 |
| BM1629 | PS-216 | *sacA*::P$_{43}$-*mKate2* (Kn) | 32 |
| PDS0036 | NCIB3610 | *amyE*::P$_{pksC}$-*yfp* (Cm) | 37 |
| BM1884 | PS-216 | *amyE*::P$_{pksC}$-*yfp* (Sp); *sacA*::P$_{43}$-*mKate2* (Kn) | This work |
| BM1070 | PS-216 | *epsA-O*::*tet* (Tc) | 75 |
| BM1310 | PS-216 | *epsA-O*::*tet* (Tc) *amyE*::P$_{hyspank}$-*mKate2* (Cm) | 76 |
| BM1899 | PS-216 | *epsA-O*::*tet* (Tc); *amyE*::P$_{pksC}$-*yfp* (Sp) | This work |
| BM1901 | PS-216 | *epsA-O*::*tet* (Tc); *amyE*::P$_{pksC}$-*yfp* (Sp); *sacA*::P$_{43}$-*mKate2* (Kn) | This work |
| BM1707 | PS-216 | Δ*srfAA* | 21 |
| BM1842 | PS-216 | Δ*srfAA*; *amyE*::P$_{hyspank}$-*mKate2* (Cm) | This work |
| PSK0178 | NCIB3610 | Δ*pks*::*spec* (Sp) | 17 |
| BM1875 | PS-216 | Δ*pks*::*spec* (Sp) | 77 |
| BM1876 | PS-216 | Δ*pks*::*spec* (Sp); *amyE*::P$_{hyspank}$-*mKate2* (Cm) | This work |
| BKE17100 | 168 | *pksC*::*ery* | 69 |
| BM1957 | PS-216 | *pksC*::*ery* | This work |
| BM1959 | PS-216 | *pksC*::*ery*; *sacA*::*pksC* (Cm) | This work |
| BM1906 | PS-216 | *epsA-O*::*tet* (Tc); Δ*pks*::*spec* (Sp); *amyE*::P$_{hyspank}$-*mKate2* (Cm) | This work |
| 168 | 168 | *trpC2* | 78 |
| BM1870 | 168 | *trpC2*; *amyE*::P$_{hyspank}$-*mKate2* (Cm) | This work |
| | | | |
| *Salmonella enterica* serovar Typhimurium | | | |
| SL1344 | NA | WT | 34 |
| SL1344 GFP | SL1344 | pFPV25 *gfpmut3* (Amp) | 72 |

[a]NA, not applicable; WT, wild type; Cm, chloramphenicol; Kn, kanamycin; Sp, spectinomycin; Tc, tetracycline; Amp, ampicillin.

subtilis BM1901 strain labeled with mKate2, the gene for which was linked to a constitutive promoter P$_{43}$ and integrated in the *sacA* locus. *S.* Typhimurium SL1344 (WT) (34, 47) was fluorescently labeled via the *gfpmut3* gene expressed from a plasmid (72, 73) (Table 1).

To prepare overnight cultures, bacterial strains were grown in tryptic soy broth (Conda, Spain) supplemented with the appropriate antibiotics at 37°C and shaken at 200 rpm for 16 h. The antibiotic concentrations in the medium were as follows: Cm 10 $\mu$g/mL, Kn 50 $\mu$g/mL, Tc 10 $\mu$g/mL, Sp 100 $\mu$g/mL, and Amp 100 $\mu$g/mL.

**Bacterial growth determination.** The growth of the *B. subtilis* PS-216 WT strain and PS-216 Δ*epsA-O* mutant was monitored by measuring the optical density at 650 nm (OD$_{650}$) on a spectrophotometer Spectroquant Prove 100 (Merck, Germany) at 30-minute intervals for up to 8 h. Briefly, overnight cultures of bacterial strains were prepared in TSB and 20-times diluted (1/20) TSB medium and incubated with shaking at 200 rpm for 16 h at 37°C. A total of 1% (V/V) of overnight cultures was transferred to fresh TSB and 1/20 TSB medium and incubated with shaking at 200 rpm and 37°C for 8 h.

**Static culture assay.** To prepare the inoculum, overnight cultures were centrifuged at 10,000 × *g* for 10 min, supernatants were discharged, and pellets were resuspended in fresh undiluted, 5-times diluted (1/5), or 1/20 TSB medium. *S.* Typhimurium suspensions were then diluted to OD$_{650}$ ~ 0.1 absorbance units (AU), while suspensions of the different *B. subtilis* strains were diluted to OD$_{650}$ ~ 0.2 AU to obtain approximately 10$^7$ cells/mL.

*S.* Typhimurium was mixed with different *B. subtilis* strains at a 1:1 ratio and properly vortexed, and 100 $\mu$L of each coculture sample was transferred into the wells of a 96-well microtiter plate with an F-bottom (Cellstar, Greiner Bio-One, Austria) and incubated further for 24 h at 37°C under static conditions. Monocultures were prepared at a 1:1 ratio, with fresh undiluted, 1/5, or 1/20 TSB medium, representing the controls. To estimate the number of cells at the beginning and the end of the experiment in monocultures and cocultures, the complete samples were disrupted by vigorous pipetting and vortexing. Samples were then diluted and plated on LB agar plates with the appropriate antibiotics to determine CFU/mL.

To calculate the inhibition of *S.* Typhimurium SL1344 GFP by the different *B. subtilis* strains, cell counts after 24 h of *S.* Typhimurium growth were used to calculate the log$_{10}$ of obtained CFU/mL values for monoculture and coculture. The inhibition of *S.* Typhimurium was then calculated by subtracting the coculture values from the monoculture values for each biological replicate (*n* = 3). Average and standard deviation are shown in Fig. 4B and Fig. S2.

**Biofilm of *S.* Typhimurium on polystyrene surface.** *S.* Typhimurium SL1344 GFP was tested for its adhesive potential to abiotic polystyrene surfaces by determining the CFU/mL of cells that adhered to the surface in monoculture and coculture with different *B. subtilis* strains. The inoculum was prepared as described above for static cultures and the protocol was carried out as described previously (21). Briefly, after 24 h of incubation at 37°C in a 96-well microtiter plate, the whole sample was removed from each well by pipetting and washed 3 times with 100 $\mu$L sterile phosphate-buffered saline (PBS). At the end, 100 $\mu$L PBS was left in the well and cells were detached from the surface by sonication in a water bath

for 10 min at low frequency (33 kHz) and 2 min at high frequency (40 kHz, 120 W power; Asonic, Slovenia) at room temperature and plated on LB agar plates with ampicillin to determine the CFU/mL of attached *S.* Typhimurium cells.

**Spatial distribution of *B. subtilis* and *S.* Typhimurium cells in biofilms.** Monocultures and cocultures of *B. subtilis* (labeled with mKate2) and *S.* Typhimurium (labeled with GFPmut3) were prepared as described previously and analyzed after 24 h of incubation using CLSM with a slightly modified protocol described by Erega et al. (21). Briefly, an inverted confocal laser scanning microscope (Axio Observer Z1, LSM800; Zeiss, Germany) was used to investigate the spatial distribution of the two species and the architecture of the mixed- and monospecies biofilms. Excitation of red fluorescent protein mKate2 was performed at 589 nm with an argon laser and emission was recorded between 580 and 700 nm. Excitation of green fluorescent protein GFPmut3 was performed at 488 nm and emission was recorded between 400 and 580 nm. The GaAsP detector gain and laser intensities were 800 V and 4.5% for mKate2 and 650 V and 0.7% for GFP, respectively. The pinhole size was 55 $\mu$m for both lasers used. For imaging the whole sample in the well in monocultures and cocultures, the thickness of one slice in the z-stack was 10 $\mu$m, while the thickness of one slice in the z-stack was 3.27 $\mu$m when determining the thickness of *S.* Typhimurium biofilms. The wells in the microtiter plate were scanned using a 20×/0.4 NA objective and the captured images were 1,180 × 1,180 pixels in size with 8-bit color depth.

To determine spatial distribution and biofilm thickness, analyses of monocultures and cocultures were performed in the same biological replicate with at least two technical repetitions. Three biological replicates were performed time-independently.

**Image analysis and processing.** For image processing, Zen 2.3 Software (Carl Zeiss) was used, and the image noise was reduced by applying a single pixel filter (threshold = 1.5). After applying the single-pixel filter, the Zen imaging software was used to assemble the 3D and orthogonal views and the results were presented as one of the most representable images. The thickness of the submerged *S.* Typhimurium biofilms was determined as the total sum of slices with visible GFP signal. To account for technical error, we deducted the first- and last-detected slices from the total sum of slices.

**Expression of P*pksC*-*yfp*.** Strains were prepared as described above for static cultures. Here, 100-$\mu$L volumes of monocultures and cocultures in TSB and 1/20 TSB medium were allocated to 3 technical replicates in the wells of a 96-well, black transparent-bottomed microtiter plate (Cellstar, Greiner Bio-One, Austria). The microtiter plate was sealed with micropore tape to minimize evaporation during incubation and then incubated statically for 24 h at 37°C in a Cytation 3 imaging reader (BioTek, USA). To monitor P*pksC*-*yfp* expression, we measured the YFP fluorescence intensity with excitation at 500 nm, emission at 530 nm, and the gain set at 100. The fluorescence intensity of mKate2 (red fluorescent protein) with excitation at 570 nm, emission at 620 nm, and the gain set at 100 was determined to monitor expression of the constitutively expressed P$_{43}$-*mKate2*. Both fluorescence intensities (YFP and mKate2) were measured every half-hour for 24 h. To calculate the promoter activity in each biological replicate, the fluorescence intensity of the unmarked monoculture was deducted from that of the monoculture carrying the fluorescent reporter YFP or mKate2, and the fluorescence intensity of the unmarked coculture was deducted from that of the coculture carrying the fluorescent reporter YFP or mKate2. Next, the mKate2 fluorescence intensity of the monoculture, representing the maximum, was set at 1. The remaining mKate2 fluorescence intensities, measured at different time points of each experimental setting, were divided by the highest mKate2 fluorescence intensity value to obtain relative constitutive fluorescence intensity for monocultures and cocultures. To compare the P*pksC* promoter activity between monocultures and cocultures, the deducted YFP fluorescence intensity was divided by the relative fluorescence intensity of constitutively expressed mKate2 to normalize the promoter activity per biomass.

To measure the expression of pksC at the single-cell level, static monocultures and cocultures were prepared in TSB and 1/20 TSB medial as described above. Three biological repeats were prepared per condition, and each biological repeat consisted of two duplicate cultures. *pksC* expression was measured after 16 h in the first duplicate and after 24 h in the second duplicate. Here, biofilms were first disrupted by vigorous pipetting and vortexing. Afterwards, 30,000 cells per sample were measured using a CytoFLEX Flow Cytometer (Beckman Coulter). Gene expression was analyzed using the CytExpert software. First, singlets were identified based on forward and side scatter. Second, the *B. subtilis* cells were identified based on the red fluorescence provided by the constitutively expressed P$_{43}$-*mKate2*. Next, the expression of the *pksC* was quantified by measuring the YFP fluorescence intensity (excitation 488 nm, emission 530 nm). To account for any background fluorescence, the measured fluorescence level was subtracted by the average yellow fluorescence intensity measured in *B. subtilis* strains which only carried P$_{43}$-*mKate2*.

To study the effect of *S.* Typhimurium monoculture and coculture conditioned medium (CM) on P*pksC*-*yfp* expression, CM was obtained by first preparing *S.* Typhimurium and *B. subtilis* strains in TSB medium as described above and transferring 2.82-mL samples into a 6-well plate (Brand, Germany) to achieve the same surface-to-volume ratio as in the 96-well microtiter plate. After 24 h of static incubation at 37°C, samples were centrifuged at 10,000 × *g* for 10 min and the isolated supernatants were passed through filters with 0.2-$\mu$m pores to obtain CM. CM was then mixed with *B. subtilis* at a 1:1 (V/V) ratio, transferred into the wells of the 96-well, black transparent-bottomed microtiter plate, and incubated further for 24 h at 37°C under static conditions in a Cytation 3 imaging reader to monitor P*pksC*-*yfp* expression.

**Statistical analysis.** All experiments were performed in at least three independent biological replicates, each represented by at least 3 technical repetitions. All the results are shown as mean values and error bars represent standard deviation. Statistical analysis of the data using a two-sample Student's *t* test was performed in OriginPro.

## SUPPLEMENTAL MATERIAL

Supplemental material is available online only.

**SUPPLEMENTAL FILE 1**, PDF file, 1.5 MB.

## ACKNOWLEDGMENTS

We acknowledge the Slovenian Research Agency (ARRS) for funding of the research program grant P4-0116, the young research program grant ARRS+ awarded to I.M.-M. and E.P., the ARRS postdoctoral grant Z4-1976 awarded to E.K., and the ARRS research grant J4-4550 awarded to I.M.-M. We also acknowledge the CELSA grant, which supported collaboration between H.S. from KU-Leuven and I.M.-M. from the University of Ljubljana and was essential to initiating the studies on interactions between *B. subtilis* and *S.* Typhimurium. We acknowledge the support of the University's infrastructural center for microscopy of biological samples located at the Biotechnical Faculty, University of Ljubljana. We also thank the National Bioresource Project (NIG, Japan): *B. subtilis* for providing us with the BKE library. H.S. acknowledges support from the KU Leuven Research Fund (C24/18/046, STG/16/022, PDM/20/123).

We would especially like to thank Paul D. Straight (Texas A&M University, Department of Biochemistry & Biophysics, USA) for his advice on complementation of the Δ*pksC* mutant and for sharing the *B. subtilis* strains. We also thank the reviewers for constructive comments which have improved the manuscript.

We have declared that we have no conflicts of interest.

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
