## [Reviewer comments · Microbiology Spectrum]

Microbiology Spectrum

Nutrient availability and biofilm polysaccharide shape the bacillaene - dependent antagonism of *Bacillus subtilis* against *Salmonella* Typhimurium

Eli Podnar, Andi Erega, Tjasa Danevčič, Eva Kovačec, Bram Lories, Hans Steenackers, and Ines Mandic-Mulec

Corresponding Author(s): Ines Mandic-Mulec, University of Ljubljana, Biotechnical Faculty

Review Timeline:

Submission Date:	May 23, 2022
Editorial Decision:	July 15, 2022
Revision Received:	October 13, 2022
Accepted:	October 18, 2022

Editor: Ilana Kolodkin-Gal

Reviewer(s): Disclosure of reviewer identity is with reference to reviewer comments included in decision letter(s). The following individuals involved in review of your submission have agreed to reveal their identity: Yunrong Chai (Reviewer #1); Giulia Barbieri (Reviewer #2)

Transaction Report:

DOI: <https://doi.org/10.1128/spectrum.01836-22>

July 15, 2022

Prof. Ines Mandic-Mulec
University of Ljubljana, Biotechnical Faculty
Food Science and technology
Večna pot 111
Ljubljana 1000
Slovenia

Re: Spectrum01836-22 (Nutrient availability and biofilm polysaccharide shape the bacillaene - dependent antagonism of *Bacillus subtilis* against *Salmonella* Typhimurium)

Dear Prof. Ines Mandic-Mulec:

Link Not Available

Sincerely,

Ilana Kolodkin-Gal

Journals Department
Reviewer comments:

Reviewer #1 (Comments for the Author):

In this study, the authors investigated the inhibitory interactions between *Bacillus subtilis* and *Salmonella* in a co-culture biofilm, and under different nutrient conditions. The authors presented evidence that the antimicrobial peptide, bacillaene, produced by *B. subtilis* plays a major role, in this inhibitory interaction. The authors further showed that nutrient conditions as well as the deletion of the *epsA-O* operon, response for biosynthesis of exopolysaccharide, a major biofilm matrix component, significantly impact this inhibitory interaction. Overall, this is an interesting study on interspecies interactions between *Bacillus* and *Salmonella*, and could provide new insights on utilizing *B. subtilis* as a probiotic. However, some of the observations and results

are preliminary, stronger and in some cases, more quantitative data and different types of assays are needed in parallel to support the claims in this study.

Major points:

Line 257 (and Fig. 2E), keep in mind that the observed surface adhesion reduction in Salmonella could simply be due to growth inhibition of Salmonella by Bacillus, thus an indirect effect. What happens if the authors try 1/5 or 1/20 TSB ? the same argument applies to lines 285-286 (and Fig. 3B).

Line 274, *B. subtilis* produces different kinds of antimicrobial compounds. It will be good if the authors can provide (briefly) some rationale here why only surfactin and bacillaene, among others, were tested.

Line 278, the conclusion here is flawed since the *sfp* gene deficiency in the strain 168 impacts production of various secondary metabolites, not just surfactin and bacillaene.

Line 293-300 (and Fig. 3C), this PpksC-yfp assay likely needs to be repeated in the test tube, under shaking conditions. The readings from static co-culture were not normalized to per cell based on description and the authors already showed the difference in cfu under mono and co-culture conditions. Alternatively, I recommend single cell resolution fluorescent images for PpksC-yfp activities and quantitative analyses to be provided in parallel.

Fig. 4B, the data presentation here is different from Figs. 1 and 3A. it will be good to keep the presentation consistent for easy comparison (I am a bit lost how to read the y-axis).

Fig. 4C, the observation that PpksC-yfp was expressed higher in the *epsA-O* mutant than in the wt, but only in 1/20 TSB is quite interesting. I would love to see strong evidence to support this.

The readings from the plate reader is not ideal in terms of quantitative measurement per cell, and growth of wt and the *epsA-O* mutant under static conditions differs. I would strongly suggest a more quantitative assay using images from the fluorescent microscopy at single-cell resolution.

Another interesting note, why would *epsA-O* mutation contributes to the Salmonella inhibition differently under nutrient-rich or poor conditions ? some discussion on this will be very helpful.

Lines 368-374, redundant from the introduction, suggest to move to introduction or shorten it.

Line 396, I doubt *eps* mutation will significantly alter nutrient utilization and spore formation in the conditions used in this study (24 hrs, static culture). If want, the authors could measure the sporulation of *B. subtilis* under these different conditions in wt and mutants.

On the other hand, it will be really interesting to learn how *epsA-O* mutation alter PpksC activity differently under different nutrient conditions, maybe for future studies.

Last, it would be interesting to show how the biofilms in TSB and diluted TSB by mono- or coculture look like, maybe as supplement (pellicle, crystal violet stain, etc).

Minor points:

Line 28, Change the words "knocking out" to "Eliminating"

Line 31, ...between exopolysaccharide production and bacillaene synthesis.

line 38, due to its antibiotic resistance and...

line 43, Hence..., this sentence reads a bit awkward, please rephrase.

Line 70, delete "when purified".

Line 74, ...and is suggested to be an important player...

Line 75, Gram-negative

Line 96, a protein can NOT be linked to a promoter, please fix it.

Line 105, Table 3 appears earlier than table 2 in the text.

Line 109, To assay bacillaene biosynthetic genes expression...,

Line 106, format error in this line (and line 324), and change "prepare" to "generate" or "resulted in"..

Line 118, change construct to generate.

Line 120, gfpmut3 gene...

Line 121, supplemented with appropriate antibiotics...

Line 123, use either abbreviations or full names in all cases, but not mixed.

Line 128, 30-min intervals for up to 8 hrs.

Line 143, at the beginning and the end of the experiment..

Line 154-156, does this protocol have a risk of significantly increasing dead cells because of sonication (12 min)? Have the authors checked that before and after sonication with, for example, live/dead staining ?

Line 159, labeled with in both places.

Line 161, described previously.

Line 275, the level of *S. Typhimurium* inhibition in coculture with either *B. subtilis* PS-216 or 168 ?

Line 317, ... is synthesized by the proteins made from the *epsA-O* operon.

Line 350, I think fig.4 is mislabeled here as fig. 5.

Reviewer #2 (Comments for the Author):

In this work, Podnar and colleagues describe the ability of *B. subtilis* to inhibit the growth and biofilm formation of *S. Typhimurium* under nutrient rich conditions. The authors demonstrate that the inhibitory effect of *B. subtilis* is dependent on its ability to produce bacillaene, whose synthesis increases during growth in rich conditions or in coculture. Notably, the activity of the PpksC promoter driving bacillaene production is influenced by the ability of *B. subtilis* to produce extracellular polysaccharides.

The work is very clear and provides an insight in the complexity of the regulation of the interaction between bacterial species.

I have some comments and suggestions:

1. Line 96: isn't the Phyperspank an IPTG inducible promoter found in the amyE integrative pDR111 vector? Why did the authors choose to use two different promoters (Phyperspank and P43?) to drive mKate2 expression? Do the two promoters display differences in their strength?
2. *B. subtilis* BM1097 and BM1629 strains both express mKate2 in the PS-216 background. Which of the two strains was used in the experiments reported in figures 1 and 2? I strongly suggest including in the figure captions the names of the strains used in the different experiments and listed in Table 1.
3. In figure 3, the addition of a Δ pks complementation mutant is required to confirm the role of bacillaene in the inhibitory activity of *B. subtilis* against *S. Typhimurium*.
4. Figure 5B. How is the the log₁₀ CFU/ml inhibition of *S. Typhimurium* in co-culture with *B. subtilis* calculated?
5. Have the authors tried to assess the effect of *S. Typhimurium* conditioned medium on PpksC activity?
6. Line 389: The following reference should be included and commented:
Schoenborn AA, Yannarell SM, Wallace ED, Clapper H, Weinstein IC, Shank EA. Defining the Expression, Production, and Signaling Roles of Specialized Metabolites during *Bacillus subtilis* Differentiation. *J Bacteriol.* 2021 Oct 25;203(22):e0033721. doi: 10.1128/JB.00337-21. Epub 2021 Aug 30. PMID: 34460312; PMCID: PMC8544424.

Staff Comments:

Preparing Revision Guidelines

To submit your modified manuscript, log onto the eJP submission site at <https://spectrum.msubmit.net/cgi-bin/main.plex>. Go to Author Tasks and click the appropriate manuscript title to begin the revision process. The information that you entered when you

first submitted the paper will be displayed. Please update the information as necessary. Here are a few examples of required updates that authors must address:

Please return the manuscript within 60 days; if you cannot complete the modification within this time period, please contact me. If you do not wish to modify the manuscript and prefer to submit it to another journal, please notify me of your decision immediately so that the manuscript may be formally withdrawn from consideration by Microbiology Spectrum.

July 15, 2022

Prof. Ines Mandic-Mulec
University of Ljubljana, Biotechnical Faculty
Food Science and technology
Večna pot 111
Ljubljana 1000
Slovenia

Re: Spectrum01836-22 (Nutrient availability and biofilm polysaccharide shape the bacillaene - dependent antagonism of *Bacillus subtilis* against *Salmonella* Typhimurium)

Dear Prof. Ines Mandic-Mulec:

<https://spectrum.msubmit.net/cgi-bin/main.plex?el=A2QF6BzWv2A3BPWF4F1A9ftdhKCB7xxQYt79mDzQGAZ>

Sincerely,

Ilana Kolodkin-Gal

Journals Department
Reviewer comments:

Reviewer #1 (Comments for the Author):

In this study, the authors investigated the inhibitory interactions between *Bacillus subtilis* and *Salmonella* in a co-culture biofilm, and under different nutrient conditions. The authors presented evidence that the antimicrobial peptide, bacillaene, produced by *B. subtilis* plays a major role, in this inhibitory interaction. The authors further showed that nutrient conditions as well as the deletion of the *epsA-O* operon, response for biosynthesis of exopolysaccharide, a major biofilm matrix component, significantly impact this inhibitory interaction. Overall, this is an interesting study on interspecies interactions between *Bacillus* and *Salmonella*, and could provide new insights on utilizing *B. subtilis* as a probiotic. However, some of the observations and results are preliminary, stronger and in some cases, more quantitative data and different types of assays are needed in parallel to support the claims in this study.

Major points:

Line 257 (and Fig. 2E), keep in mind that the observed surface adhesion reduction in *Salmonella* could simply be due to growth inhibition of *Salmonella* by *Bacillus*, thus an indirect effect. What happens if the authors try 1/5 or 1/20 TSB ? the same argument applies to lines 285-286 (and Fig. 3B).

The reviewer is correct - the decrease in the number of adhered cells is most probably due to the growth inhibition of *Salmonella* because the total number of cells correlates with the adhered cells. We emphasized this point additionally and we therefore modified the text accordingly:

“This assay confirmed that the *B. subtilis* strain was able to significantly reduce *S. Typhimurium* adhesion, probably due to growth inhibition (Fig. 2E).” (p. 6, l. 131-132)

“In contrast, the *B. subtilis* PS-216 Δpks mutant, that does not produce bacillaene, completely lost its inhibitory effect against the pathogen (Fig. 3A) and consequently its ability to lower the adhesion of *Salmonella* to the polystyrene surface (Fig. 3B).” (p.7, l. 144-146).

Because we did not observe significant growth inhibition by PS-216 in TSB 1/5 and 1/20 media we have not tested the effect of *B. subtilis* on adhesion in these media.

Line 274, *B. subtilis* produces different kinds of antimicrobial compounds. It will be good if the authors can provide (briefly) some rationale here why only surfactin and bacillaene, among others, were tested.

Thank you for the comment. We first tested surfactin, which has been previously associated with inhibition of *Salmonella* biofilm, but we have not observed any loss of inhibition. In contrast, the *pks* mutant completely lost the inhibitory activity against *Salmonella*. Therefore, we concluded that bacillaene is the key antagonistic factor.

Line 278, the conclusion here is flawed since the *sfp* gene deficiency in the strain 168 impacts production of various secondary metabolites, not just surfactin and bacillaene.

The reviewer here accurately emphasizes that the *sfp* gene deficiency in strain 168 impacts the production of different antimicrobial compounds, therefore, we additionally tested mutants defective in plipastatin synthesis (*ppsB*). Plipastatin synthesis is known to be negatively affected by

the *sfp* gene mutation (Tsuge *et al.*, 1999, *Antimicrob Agents Chemother* 43:2183–2192). In addition, we also tested the effect of bacilysin on *Salmonella*. Both the plipastatin and bacilysin mutants inhibited *Salmonella* growth, thus further confirming that bacillaene is the key antagonistic factor. We modified the text in the revised manuscript (p.6 and 7, l. 135-146) and the new results are included in Fig 3A.

Line 293-300 (and Fig. 3C), this PpksC-yfp assay likely needs to be repeated in the test tube, under shaking conditions. The readings from static co-culture were not normalized to per cell based on description and the authors already showed the difference in cfu under mono and co-culture conditions. Alternatively, I recommend single cell resolution fluorescent images for PpksC-yfp activities and quantitative analyses to be provided in parallel.

Repeating the P_{pksC} -yfp assay in shaking conditions has the drawback that it is not well aligned with our goal of studying interactions during biofilm growth, which is more relevant in regard to pathogen control. All the other experiments were therefore performed in static conditions. Besides, data from the static coculture were normalized to the expression of the constitutive promoter P_{43} -mKate2, which represents a measure for total biomass as previously described (Spacapan *et al.*, 2020. *Microorganisms*, 8: 1131). Therefore, our data points were normalized per biomass. However, to further support our findings, and as suggested by the reviewer, we have now performed quantitative analysis of P_{pksC} -yfp activity per cell using flow cytometry. The new results are included in the revised manuscript as Fig. S3. We modified the text according to the new data (p. 7 and 8, l. 160-165 and 166-168), which is consistent with the bulk data obtained in the microplate reader.

Fig. 4B, the data presentation here is different from Figs. 1 and 3A. it will be good to keep the presentation consistent for easy comparison (I am a bit lost how to read the y-axis).

While in Fig. 1 and Fig. 3A we showed CFU counts of monoculture and coculture, in Fig. 4B we presented data as Log of growth inhibition of *Salmonella* between three strains and two media. Although the presentation is not the same in all the figures, we believe that the point is still clear in Fig. 4B as we wanted to emphasize here the level of growth inhibition. To make this clearer, we have now additionally explained how the calculations were performed in the methods section (p. 14, l. 319-323).

Fig. 4C, the observation that PpksC-yfp was expressed higher in the *epsA-O* mutant than in the wt, but only in 1/20 TSB is quite interesting. I would love to see strong evidence to support this. The readings from the plate reader is not ideal in terms of quantitative measurement per cell, and growth of wt and the *epsA-O* mutant under static conditions differs. I would strongly suggest a more quantitative assay using images from the fluorescent microscopy at single-cell resolution. Another interesting note, why would *epsA-O* mutation contributes to the *Salmonella* inhibition differently under nutrient-rich or poor conditions? some discussion on this will be very helpful.

We thank the reviewer for the comments.

We have now performed additional quantitative measurements to address the role of $\Delta epsA-O$ on P_{pksC} -yfp activity per cell by flow cytometry. The results are consistent with bulk measurements in nutrient-rich conditions – again we did not observe a significant increase in P_{pksC} -yfp activity in the $\Delta epsA-O$ mutant at single cell level. However, in nutrient-depleted conditions (1/20 TSB medium) the P_{pksC} activity was similar to that in rich conditions at the 24-hour time point and even slightly lower at the 16-hour time point. The new results are included in the revised manuscript in Fig. S6 and we modified our conclusions accordingly. We now state: “After 15 hours in nutrient-depleted conditions, the bulk P_{pksC} activity in monoculture conditions strongly declined, whereas it remained

stable in coculture (Fig. 4C). However, these observations cannot be validated by single cell measurements (Fig. S6)". (p. 9, l. 201-203).

We also further investigated the growth rates of the mutant and the wild type. The wild type and the $\Delta epsA-O$ mutant grow with very similar rates in the 1/20 medium in our system, regardless of whether the cultures were grown under static or shaking conditions (See Fig. S5 and the figure below).

Why $epsA-O$ mutation contributes differently to the *Salmonella* inhibition in different conditions is discussed in the results (p. 9, l. 198-208) and in the discussion (p. 10, l. 226-231) sections of the revised manuscript. In our opinion the difference in the inhibition is not due to a higher growth rate of the $\Delta epsA-O$ mutant because the growth rates of the wild type and the mutant are similar. We therefore speculate that the higher inhibitory activity of the mutant in 1/20 TSB medium is likely due to its lower sporulation frequency, which may prolong the production of bacillaene. Lower sporulation frequency of the mutant has been previously reported as cited in references Vlamakis et al, 2008. *Genes and development* 22:945–953; Aguilar et al., 2010. *mBio* 1(1):e00035-10 (references #65, 66).

As suggested by the reviewer we have also compared the sporulation frequency of the PS-216 wild type and the $\Delta epsA-O$ mutant and our results confirm previous findings as shown in Fig. S7 (also see below).

Figure 1. Growth curves of *B. subtilis* PS-216 WT strain (BM1097) and *B. subtilis* PS-216 $\Delta epsA-O$ mutant (BM1310) in 1/20 TSB medium. Bacterial strains were grown in monoculture in 1/20 TSB media for 16 hours statically at 37 °C. Measurements were performed every half an hour and means with standard deviations of three biological replicates are shown.

Lines 368-374, redundant from the introduction, suggest to move to introduction or shorten it.

The text has been removed from the revised manuscript.

Line 396, I doubt eps mutation will significantly alter nutrient utilization and spore formation in the conditions used in this study (24 hrs, static culture). If want, the authors could measure the sporulation of *B. subtilis* under these different conditions in wt and mutants.

Thank you, we have tested the sporulation frequency in the wild type strain and $\Delta epsA-O$ mutant in 1/20 TSB medium after 24 hours of incubation and provide the Fig. S7. It confirmed prior reports showing that matrix mutants are delayed in sporulation (Aguilar *et al.*, 2010. mBio 1(1):e00035-10).

Figure S7: The sporulation frequency of *B. subtilis* PS-216 WT strain and PS-216 $\Delta epsA-O$ strains after 24 hours of incubation in 1/20 TSB medium.

On the other hand, it will be really interesting to learn how *epsA-O* mutation alter PpkcC activity differently under different nutrient conditions, maybe for future studies.

The data for *epsA-O* impact on P_{pkcC} activity under nutrient rich and nutrient-depleted conditions were already included in the manuscript in Fig. 4C.

Last, it would be interesting to show how the biofilms in TSB and diluted TSB by mono- or coculture look like, maybe as supplement (pellicle, crystal violet stain, etc).

Thank you for your advice. We have added images of the biofilms in different growth conditions in Fig. S1 as suggested and modified the text in the revised manuscript (p. 5, l.116 and p. 6, l.181-184).

Minor points:

Line 28, Change the words "knocking out" to "Eliminating"

We have corrected the text in the revised manuscript.

Line 31, ...between exopolysaccharide production and bacillaene synthesis.

We have corrected the text in the revised manuscript.

line 38, due to its antibiotic resistance and...

We have corrected the text in the revised manuscript.

line 43, Hence..., this sentence reads a bit awkward, please rephrase.

We have rephrased the sentence in the revised manuscript.

Line 70, delete "when purified".

We have deleted the text in the revised manuscript as suggested.

Line 74, ...and is suggested to be an important player...

We have corrected the text in the revised manuscript.

Line 75, Gram-negative

We have corrected the text in the revised manuscript.

Line 96, a protein can NOT be linked to a promoter, please fix it.

We have corrected the text in the revised manuscript.

Line 105, Table 3 appears earlier than table 2 in the text.

We have changed the order of the table 2 and table 3. Table 3 became table 2 and opposite in the revised manuscript.

Line 109, To assay bacillaene biosynthetic genes expression...,

We have corrected the text in the revised manuscript.

Line 106, format error in this line (and line 324), and change "prepare" to "generate" or "resulted in"..

We have corrected the formatting and changed the text as suggested in the revised manuscript.

Line 118, change construct to generate.

We have changed the text in the revised manuscript.

Line 120, gfpmut3 gene...

We have corrected the text in the revised manuscript.

Line 121, supplemented with appropriate antibiotics...

We have corrected the text in the revised manuscript.

Line 123, use either abbreviations or full names in all cases, but not mixed.

We have corrected the text in the revised manuscript.

Line 128, 30-min intervals for up to 8 hrs.

We have corrected the text in the revised manuscript.

Line 143, at the beginning and the end of the experiment.

We have corrected the text in the revised manuscript.

Line 154-156, does this protocol have a risk of significantly increasing dead cells because of sonication (12 min)? Have the authors checked that before and after sonication with, for example, live/dead staining ?

When optimising the protocol, the samples going through the sonification process have been checked before and after sonication using live/dead staining. There was no significant increase in the number of dead cells due to the prolonged sonication.

Line 159, labeled with in both places.

We have corrected the mistake in the revised manuscript.

Line 161, described previously.

We have corrected the text in the revised manuscript.

Line 275, the level of *S. Typhimurium* inhibition in coculture with either *B. subtilis* PS-216 or 168 ?

We have corrected the text in the revised manuscript.

Line 317, ... is synthesized by the proteins made from the *epsA-O* operon.

We have corrected the text as suggested in the revised manuscript.

Line 350, I think fig.4 is mislabeled here as fig. 5.

Thank you for notice the mislabelling. We have corrected the mistake in the revised manuscript.

Reviewer #2 (Comments for the Author):

In this work, Podnar and colleagues describe the ability of *B. subtilis* to inhibit the growth and biofilm formation of *S. Typhimurium* under nutrient-rich conditions. The authors demonstrate that the inhibitory effect of *B. subtilis* is dependent on its ability to produce bacillaene, whose synthesis increases during growth in rich conditions or in coculture. Notably, the activity of the *PpksC* promoter driving bacillaene production is influenced by the ability of *B. subtilis* to produce extracellular polysaccharides.

The work is very clear and provides an insight in the complexity of the regulation of the interaction between bacterial species.

I have some comments and suggestions:

1. Line 96: isn't the Phyperspank an IPTG inducible promoter found in the amyE integrative pDR111 vector? Why did the authors choose to use two different promoters (Phyperspank and P43?) to drive mKate2 expression? Do the two promoters display differences in their strength?

This is a good point. In our study we used the constitutive promoter $P_{hyperspank}$ constructed previously (Van Gestel *et al.*, 2014. ISME Journal, 8: 2069–2079). They constructed it using $P_{hyperspank}$ from the pDR111 vector. The plasmid lacks the *lacI* gene encoding the repressor and therefore $P_{hyperspank}$ is constitutive rather than IPTG-inducible in this construct. In some publications, this promoter is named $P_{hyperspank}$ whereas in others it is designated as $P_{hyspank}$. To avoid confusion, we changed the name of the promoter to $P_{hyspank}$ in the revised manuscript and Table 1.

We used two different promoters for convenience during mutant construction. Although the promoters show slightly different strength, they are both constitutive. However, it is important to note that we have not directly quantitatively compared strains with two different promoters. We used promoter $P_{hyspank}$ in experiments for qualitative measurements of biofilms using microscopy (for visualization) and promoter P_{43} in experiments for measuring fluorescence intensity and consequently P_{pksC} activity per cell.

2. *B. subtilis* BM1097 and BM1629 strains both express mKate2 in the PS-216 background. Which of the two strains was used in the experiments reported in figures 1 and 2? I strongly suggest including in the figure captions the names of the strains used in the different experiments and listed in Table 1.

We have changed the text as requested. In Fig. 1, Fig. 2, Fig. 3A, Fig. 4B, Fig. S1, Fig. S2, Fig. S5, Fig. S7 we used *B. subtilis* strain BM1097, while for the experiments that measure P_{pksC} activity (Fig. 3C, Fig. S3, and Fig. S4) *B. subtilis* strain BM1629 was used as explained above.

3. In figure 3, the addition of a Δpks complementation mutant is required to confirm the role of bacillaene in the inhibitory activity of *B. subtilis* against *S. Typhimurium*.

In our study, a *pks* mutant with almost the whole cluster (from *pksB* to *pksR* – approximately 75,8 kb) replaced by a spectinomycin cassette was used (the template strain from Paul Straight) (Straight *et al.*, 2007. PNAS, 104, 1: 305-310). We agree with the reviewer's comment and did the *pks* complementation as suggested. As the *pksC* gene is the key factor in bacillaene synthesis and as the P_{pksC} promoter, which controls expression of the *pks* gene cluster (Vargas-Bautista *et al.*, 2014. J. Bacteriol., 196, 4: 717-728) was used in our work as the main reporter, we decided to complement only the *pksC* gene. We first prepared the $\Delta pksC$ mutant and showed that it has a comparable phenotype to the Δpks mutant lacking the genes from *pksB* to *pksR* (Fig. S2) and then we complemented this mutation in PS-216 *pksC::ery* with the *pksC* gene integrated into the *sacA* locus (*sacA::pksC* (Cm)). The detailed mutant's preparation is described in the materials and methods section (p. 12, l. 271-281), changes were also made in results section (p. 6, l. 146-149) and the results are added in the Fig. S2.

4. Figure 5B. How is the the log10 CFU/ml inhibition of *S. Typhimurium* in co-culture with *B subtilis* calculated?

Cell counts after 24 hours of *S. Typhimurium* incubation were used to calculate CFU/mL and Log₁₀ of obtained CFU/mL values for monoculture and coculture. The inhibition of *S. Typhimurium* was then calculated by subtracting the coculture values from the monoculture values for each biological replicate (n=3). Averages and standard deviations are shown in Fig. 4B. The explanation of the calculation was added in materials and methods section (p. 14, l. 319-323).

5. Have the authors tried to assess the effect of *S. Typhimurium* conditioned medium on PpksC activity?

Thank you for this interesting point. Yes, we also tested the effect of *S. Typhimurium* conditioned medium on P_{pksC} activity in the BM1629 strain in TSB medium. The results showed no increase in P_{pksC} activity either in conditioned medium collected from *S. Typhimurium* monoculture or *S. Typhimurium* and *B. subtilis* coculture. We have included these data in the revised manuscript as Fig. S4 and stated this in the results section (p. 8, l. 168-171).

6. Line 389: The following reference should be included and commented:
Schoenborn AA, Yannarell SM, Wallace ED, Clapper H, Weinstein IC, Shank EA. Defining the Expression, Production, and Signaling Roles of Specialized Metabolites during *Bacillus subtilis* Differentiation. *J Bacteriol.* 2021 Oct 25;203(22):e0033721. doi: 10.1128/JB.00337-21. Epub 2021 Aug 30. PMID: 34460312; PMCID: PMC8544424.

We agree that this is an important reference. The proposed reference (#64) has already been included in the manuscript (now p. 10., l. 226).

Staff Comments:

Preparing Revision Guidelines

For complete guidelines on revision requirements, please see the journal Submission and Review Process requirements at <https://journals.asm.org/journal/Spectrum/submission-review-process>.

Submissions of a paper that does not conform to Microbiology Spectrum guidelines will delay acceptance of your manuscript. "

Please return the manuscript within 60 days; if you cannot complete the modification within this time period, please contact me. If you do not wish to modify the manuscript and prefer to submit it to another journal, please notify me of your decision immediately so that the manuscript may be formally withdrawn from consideration by Microbiology Spectrum.

October 18, 2022

Prof. Ines Mandic-Mulec
University of Ljubljana, Biotechnical Faculty
Department of Microbiology
Večna pot 111
Ljubljana 1000
Slovenia

Re: Spectrum01836-22R1 (Nutrient availability and biofilm polysaccharide shape the bacillaene - dependent antagonism of *Bacillus subtilis* against *Salmonella* Typhimurium)

Dear Prof. Ines Mandic-Mulec:

Your manuscript has been accepted, and I am forwarding it to the ASM Journals Department for publication. You will be notified when your proofs are ready to be viewed.

Sincerely,

Ilana Kolodkin-Gal
Editor, Microbiology Spectrum
